# Pandora’s Box: Disseminated Coccidioidomycosis Associated with Self-Medication with an Unregulated Potent Corticosteroid Acquired in Mexico

**DOI:** 10.3390/tropicalmed6040207

**Published:** 2021-12-05

**Authors:** Alejandro Jose Coba, Patricia K. Sallee, Danielle O. Dixon, Rahaf Alkhateb, Gregory M. Anstead

**Affiliations:** 1San Antonio Infectious Diseases Associates, 8042 Wurzbach Road, San Antonio, TX 78229, USA; jacobaes@gmail.com; 2Division of Infectious Diseases, Department of Medicine, University of Texas Health, San Antonio, 7703 Floyd Curl Dr, San Antonio, TX 78229, USA; sallee@uthscsa.edu (P.K.S.); dixond1@uthscsa.edu (D.O.D.); 3Department of Pathology and Laboratory Medicine, University of Texas Health, San Antonio, 7703 Floyd Curl Dr, San Antonio, TX 78229, USA; Alkhateb@uthscsa.edu; 4Division of Infectious Diseases, Medical Service, South Texas Veterans Healthcare System, 7400 Merton Minter Blvd, San Antonio, TX 78229, USA

**Keywords:** disseminated coccidioidomycosis, corticosteroid, cutaneous coccidioidomycosis, coccidioidal synovitis, pulmonary coccidioidomycosis, coccidioidal chorioretinitis

## Abstract

Coccidioidomycosis (CM), caused by the dimorphic fungi *Coccidioides immitis* and *C. posadasii*, typically presents as acute or chronic pulmonary disease. However, disseminated disease occurs in about 1% of patients. Disseminated CM may affect multiple organ systems, including cutaneous, osteoarticular, and central nervous system sites. Here, we present a case of disseminated CM in a patient from a border city in Texas. The patient had a history of uncontrolled diabetes mellitus and was also taking an over-the-counter medication acquired in Mexico that contained a potent corticosteroid. The patient presented with seizures and was found to have a brain infarct, cavitary lung lesions, synovitis of the knee, multiple skin lesions, and chorioretinitis. The patient had a very high complement fixation titer for *Coccidioides*; fungal spherules were seen in a skin biopsy specimen, and *Coccidioides* grew in culture from a sample of synovial fluid and the skin biopsy specimen. This case illustrates the dissemination potential of *Coccidioides*, the danger of unregulated pharmaceuticals, the importance of thorough history taking, and recognizing risk factors that contribute to disseminated CM.

## 1. Introduction

Coccidioidomycosis (CM) is caused by dimorphic fungi of the genus *Coccidioides (C. immitis* (mostly California isolates) and *C. posadasii* (other areas)). It typically presents as acute and chronic pulmonary disease. However, the infection may disseminate widely, especially to cutaneous, osteoarticular, and central nervous system sites. Coccidioidomycosis is a significant health problem in the southwestern USA and Latin America, with 350,000 cases of acute infections occurring annually in the United States alone [1,2,3]. About 1% of CM cases result in dissemination, and one-third of these cases are fatal [2].

The aim of the current study is to report a case of widely disseminated CM that highlights the danger of self-medication with an unregulated pharmaceutical, the importance of thorough history taking, and recognizing risk factors that contribute to disseminated CM. The patient presented with seizures and was found to have a brain infarct, cavitary lung lesions, synovitis of the knee, multiple skin lesions, and chorioretinitis. Coccidioidomycosis was confirmed by serologic and histopathologic methods and by fungal culture. The risk factors for dissemination in this patient were uncontrolled diabetes mellitus and his use of an over-the-counter medication from Mexico that contained a potent corticosteroid.

## 2. Case

A 50-year-old Hispanic man was transferred to University Hospital in San Antonio, Texas, from an emergency department in Eagle Pass, Texas, for evaluation of transient loss of consciousness with witnessed seizure activity. Prior to transfer, the patient underwent computerized tomography of the head, cervical spine, chest, abdomen, and pelvis that were only significant for a 1.9 cm left lower lung lobe cavity with an air-fluid level (Figure 1). Upon hospital admission, the patient was empirically treated for necrotizing pneumonia with ampicillin–sulbactam. Magnetic resonance imaging of the brain showed possible recent lacunar infarct (Figure 2).

A computerized tomographic angiogram (CTA) of the head and neck revealed no occlusions, aneurysm, stenosis, or vascular malformations. The CTA of the head and neck disclosed a 5 mm right apical cavitary lesion that was previously appeared as a ground-glass opacity on prior imaging. The patient reported chronic cough with intermittent sputum production of several months’ duration, along with skin lesions on the left thumb, the nape of the neck, the left side of the face, and the feet (Figure 3).

A shave biopsy of his right arm lesion showed granulomatous dermatitis with verrucous and pseudoepitheliomatous squamous proliferation; a periodic acid-Schiff stain revealed fungal organisms consistent with *Coccidioides* sp. (Figure 4, Figure 5 and Figure 6). Cultures from skin biopsy were positive for *Coccidioides* sp. as well.

He also reported moderate right knee pain with associated swelling that had been present for several months. He denied any preceding trauma to the knee. He also denied weight loss, fever, chills, or night sweats. Past medical history included diabetes mellitus; he was previously on metformin but discontinued it due to gastrointestinal intolerance. He also reported persistent joint pain of the bilateral hands, wrists, and elbows, for which he was taking an over-the-counter arthritis medication that he had obtained across the border in Piedras Negras, Mexico, called Ardosons^®^ (indomethacin (25 mg), betamethasone (0.75 mg), and methocarbamol (215 mg) in each tablet). He had been taking this medication intermittently for more than a year. Vital signs on admission were within normal limits. A physical exam revealed the aforementioned verrucous plaques and nodules (Figure 3). His right knee was swollen and tender to palpation; no erythema or decreased range of motion was noted. A complete blood count showed white blood cell count 8.9 K/μL (reference range (RR) 3.4–10.4), hemoglobin 12.1 g/dL (RR 12.8–17.1), and platelets 383 K/μL (RR 140–377). A comprehensive metabolic panel was within normal limits. The hemoglobin A1c was elevated at 12.5% (RR 4.0–6.0). The patient’s C-reactive protein was elevated at 97.9 mg/L (RR: <10). Further evaluation showed the following negative results: sputum acid-fast stain and culture; human immunodeficiency virus serologic testing; *Histoplasma* urine antigen; *Cryptococcus* serum antigen; and an interferon gamma release assay for tuberculosis. *Coccidioides* IgM and IgG levels by ELISA were 0.2 and 7.9, respectively (RR: ≤0.9, negative; 1.0–1.4, equivocal; ≥1.5, positive). *Coccidioides* antibody by immunodiffusion was detected, and the complement fixation titer was 1:256 (RR < 1:2).

A computerized tomograph of the right knee showed no acute osseous defects and a moderate suprapatellar joint effusion (Figure 7). Synovial fluid aspirated from the right knee showed: no crystals; 20,000 RBC/mm^3^; and 26,100/mm^3^ total nucleated cells with 84% neutrophils, 11% lymphocytes, and 5% monocytes. A right knee arthrotomy with drainage and irrigation was performed; serosanguineous cloudy fluid was obtained, which grew *Coccidioides* (Figure 8).

Given the initial presentation of loss of consciousness with seizure activity, a lumbar puncture was performed to assess CNS dissemination. The cerebrospinal fluid (CSF) showed protein 46 mg/dL (RR: ≤45), glucose 70 mg/dL, total nucleated cells <5/mm^3^ (81% lymphocytes), and red blood cells 2/mm^3^. The *Coccidioides* CSF IgM and IgG levels were 0.0 and 3.8, respectively (RR: ≤0.2, negative; ≥0.3, positive). *Coccidioides immitis* CSF antibody by ID and CF were <1:2 (not detected). The CSF fungal culture showed no growth. Although the patient denied ocular symptoms, given the widely disseminated disease, the ophthalmology service was consulted for a retinal exam, which showed findings indicative of chorioretinitis (Figure 9). There was no vitritis or uveitis. The patient was placed on fluconazole 800 mg daily.

The patient was last followed up in the telemedicine outpatient infectious diseases clinic ten weeks after discharge. He continued to have mild diffuse joint pain but was overall improved. He denied headache, cough, and dyspnea. He remained uninsured and had difficulty affording his fluconazole, so the dose was decreased from 800 mg orally per day to 600 mg daily. An X-ray of his knee performed five weeks after his knee surgery showed a persistent, moderate right knee effusion without underlying bone or joint abnormality. He noted that he was no longer taking any medications from Mexico. The current therapeutic plan is to continue fluconazole indefinitely due to the probable CNS infection.

## 3. Discussion

### 3.1. Dissemination in Coccidioidomycosis and in This Patient

This patient showed probable CM of the lung and multisite dissemination to the knee joint, multiple skin locations, the brain, and the chorioretina. Disseminated CM is more likely when the CF titer is ≥1:16 [4], and this patient had a CF titer of 1:256. Patients with CM with CF titers of ≥1:256 are also more likely to suffer relapses [5]. About 1% of CM cases result in disseminated disease, and about one-third of these cases result in death. Immunocompromised persons are at high risk for fatal disseminated CM; the crude mortality rate is about 50% for CM patients that are immunocompromised by HIV, cancer, organ transplantation, anti-rejection medications, anti-inflammatory biological treatments, or chemotherapy. However, the risk of death is lower for patients who are able to stop their immunosuppressive medications, which applies to this patient [2].

The specific risk factors for increased severity of infection and dissemination in this patient were corticosteroid use and diabetes mellitus. Corticosteroids adversely affect cell-mediated immunity, which is vital to the control of CM. Circulating T cells can be rapidly depleted in the setting of high-dose corticosteroid use, mainly due to the inhibition of growth cytokines, impaired release from lymphoid tissues, and the induction of apoptosis [6,7]. Our patient was taking a medication called Ardosons, which contains 0.75 mg betamethasone in each tablet, for more than a year. Betamethasone has 7.5-X the glucocorticoid potency of prednisone and has a duration of action of 36–72 h, compared to 12–36 h for prednisone. The immunosuppressant effects of corticosteroids are determined by both their current dosing and the cumulative dosage that has been received [8].

This patient also had uncontrolled diabetes (A1c = 12.5%), which is a known risk factor for increased severity of CM, with increasing likelihood of pulmonary cavitation, dissemination, and delayed resolution of infection [9]. In addition to its direct impact on cell-mediated immunity, betamethasone may have also had an indirect detrimental effect by exacerbating hyperglycemia, a well-known adverse drug reaction of corticosteroids [8].

### 3.2. Pulmonary Manifestations of Coccidioidomycosis and in This Patient

With inhalation of coccidioidal arthroconidia, up to 60% of people develop an asymptomatic or mild respiratory illness [4]. Symptomatic patients often present with fever, cough, and pleuritic chest pain and are often mistaken for having community-acquired pneumonia. The typical incubation period is 1–3 weeks after exposure to the arthroconidia. Chest imaging usually shows segmental or lobar consolidations, and hilar or mediastinal lymphadenopathy may be present. Patients may also develop pleural effusions, which are often exudative, with lymphocytic predominance, and eosinophilia may be present. Patients can also present with diffuse pneumonia, usually in the setting of a high inoculum exposure or immunosuppression. These patients have more severe symptoms, are ill-appearing, and can progress to acute respiratory distress syndrome. A subset of patients will develop chronic pulmonary CM with prolonged fever, cough, and weight loss. These patients may advance to pulmonary fibrosis and cavitation [10]. About 5% of patients with resolution of primary pneumonic infiltrates may develop a pulmonary nodule or cavity [4]. Our patient presented with a chronic cough with intermittent sputum production, ongoing for months, and was found to have a 1.9 cm pulmonary cavity in the left lower lobe as well as a 5 mm right apical cavity. Cavities smaller than 2.5 cm in diameter tend to resolve spontaneously over a year, but cavities larger than 5 cm persist [10]. A small proportion of non-resolving cavities may result in complications (hemorrhage, pneumothorax, or empyema) [11].

### 3.3. Central Nervous System Manifestations of Coccidioidomycosis and in This Patient

The most common central nervous system (CNS) presentation of CM is basilar meningitis, which occurs in about 50% of cases of disseminated CM. Other CNS complications include hydrocephalus, vasculitic infarction, mass lesions, abscesses, cranial neuropathy, and arachnoiditis. Based on radiographic findings, the frequency of vasculitic infarction in coccidioidal meningitis is as high as 40%. Clinical presentations of vasculitic infarction (e.g., aphasia, hemianopsia, hemiparesis) occur in about 10% of meningitis cases [12,13,14]. Cerebrovascular complications, such as ischemic strokes, intracranial hemorrhage, aneurysms, and vasospasm, are the leading cause of death in patients with CM [15]. The lacunar infarct observed in this patient is consistent with brain vasculitis due to disseminated CM [16,17]. Although the CSF cultures showed no growth in this patient, cultures are positive in less than 50% of cases of CNS CM, but the CSF IgG is usually positive [10], as in this case.

### 3.4. Osteoarticular Manifestations of Coccidioidomycosis and in This Patient

Skeletal involvement occurs in 20–56% of cases of disseminated CM. The axial skeleton is the most common site of osseous dissemination, but any bone or joint may be affected [18]. Coccidioidal arthritis usually results from an extension of osteomyelitis from adjacent bones and is thus typically associated with periarticular bony erosion. The absence of osseous destruction in this case indicates that that the knee joint infection was localized to the synovium, i.e., synovitis, a less common osteoarticular manifestation of CM [19]. For coccidioidal synovitis, the knee is the most frequently involved joint [20].

### 3.5. Skin Manifestations of Coccidioidomycosis and in This Patient

Skin lesions in CM take the form of papules, nodules, abscesses, verrucous plaques, or ulcers, with variable levels of granuloma formation depending on the immunocompetence of the patient [10]. In one series of 201 patients with disseminated CM, skin involvement was present in 15% [21]. In this patient, a histopathologic specimen from the shave biopsy from a verrucous lesion on the right arm showed granulomatous dermatitis with verrucous and pseudoepitheliomatous squamous proliferation; a periodic acid-Schiff stain revealed *Coccidioides* spherules (Figure 5 and Figure 6). The histopathologic specimen revealed multinucleated giant cells with internalized spherules. Pseudoepitheliomatous squamous proliferation occurs in response to infectious, inflammatory, and neoplastic conditions [22] and has been previously reported in CM [23,24]. Pseudoepitheliomatous squamous proliferation presents clinically as plaques or nodules with variable scaling and crusting. The excessive epithelial hyperplasia causes the skin lesion to assume a verrucous appearance [25].

### 3.6. Ocular Manifestations of Coccidioidomycosis and in This Patient

Ocular involvement in CM has been seldom reported in the literature but may affect any substructure of the eye. Clinical findings in ocular CM may range from asymptomatic focal chorioretinitis to chronic granulomatous iridocyclitis to fulminating granulomatous panendophthalmitis [26]. Ophthalmological evaluation is not regarded as standard of care in patients diagnosed with CM without ocular complaints. However, dissemination of *Coccidioides* to the eye can be asymptomatic, as in our patient. The prevalence of ocular CM is not known. A retrospective study conducted in Arizona found that five out of 54 patients with documented past infection with CM had characteristic inactive peripheral chorioretinal scars despite no prior documentation of ocular involvement [27]. Treatment of symptomatic ocular CM usually includes systemic treatment with amphotericin B with the addition of intravitreal amphotericin if severe symptoms are present, with eventual de-escalation to oral azoles [28]. The use of intravitreal amphotericin falls to the discretion of the attending ophthalmologist based on the severity of infection. Other treatment options include oral fluconazole, oral voriconazole, and intravitreal voriconazole. Unfortunately, some cases of severe ocular CM are refractory to medical treatment [29,30,31] and ultimately require enucleation [26,28,32]. The appropriate duration of treatment for ocular CM is unknown. For the patient described herein, in the right eye, there was a flat yellowish lesion along the inferior arcade and several pinpoint lesions just inferior to the fovea suggestive of chorioretinitis. Neither eye displayed uveitis, vitritis, or disc changes. Because the patient reported no visual complaints, intraocular therapy was not deemed necessary, and he will receive lifetime azole therapy due to concurrent CNS involvement [33].

### 3.7. The Dangers of Self-Medication

Self-medication is defined as the use of medicines by individuals to treat self-recognized conditions or symptoms. The potential risks of self-medication include incorrect self-diagnosis; delays in seeking appropriate medical care; adverse drug reactions; dangerous drug interactions; incorrect manner of administration, dosage, or choice of therapy; masking of a significant medical condition; and possible dependency and abuse [34]. In this case, the patient self-diagnosed himself with arthritis. Because he had no medical insurance and had a low-wage job, he decided to self-medicate with Ardosons pills, based on the advice of a family member. The patient lived in an American city on the Mexican border, and this product was readily available in Mexico. Ardosons is advertised as a treatment for rheumatoid arthritis, osteoarthritis, and other inflammatory musculoskeletal conditions (gout, synovitis, and tendonitis). The danger of American patients seeking drugs for self-medication in Mexico is an ongoing problem and was described in an article in the *Los Angeles Times* in 1999. In that article, a drug called Artridol, with ingredients identical to Ardosons, was incriminated as dangerous because of its potential to cause Cushing’s syndrome and gastrointestinal bleeding [35]. In the era of e-commerce and social media, the risk of self-medication has exploded, as the COVID-19 pandemic has amply demonstrated [36]. A Google search using the term “Ardosons” yielded several online pharmacies located in Mexico that have the product available for purchase. This case highlights the importance of a thorough historical interview that includes all medications that a patient may be taking outside of standard medical care.

## 4. Conclusions

Coccidiodomycosis (CM), caused by the dimorphic fungi *Coccidioides immitis* and *C. posadasii*, typically presents as acute or chronic pulmonary disease. However, disseminated disease occurs in about 1% of patients. Disseminated CM may affect multiple organ systems, especially cutaneous, osteoarticular, and central nervous system sites. Here, we present a case of disseminated CM in a patient from a border city in Texas. The patient had a history of uncontrolled diabetes mellitus and was also taking an over-the-counter medication from Mexico that contained a potent corticosteroid. He presented with a stroke and was found to have cavitary lung lesions, synovitis of the knee, multiple skin lesions, and chorioretinitis. The patient had a very high complement fixation titer for *Coccidioides*; *Coccidioides* was also seen in a skin biopsy specimen, and it grew in culture from the biopsy specimen and a sample of synovial fluid. This case demonstrates the potential danger of unregulated pharmaceuticals, the importance of thorough history taking, and recognizing the risk factors that contribute to disseminated CM.

## Figures and Tables

**Figure 1 tropicalmed-06-00207-f001:**
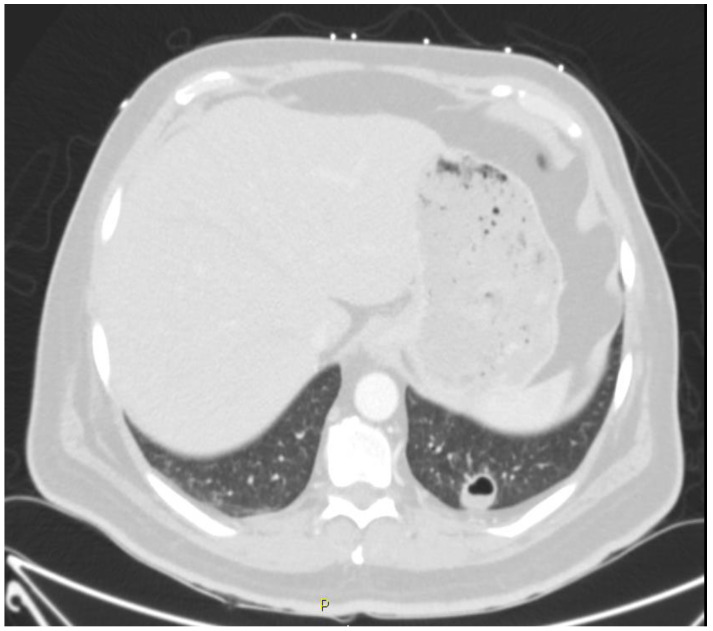
Computerized tomograph of the chest showing a 1.7 cm × 2 cm × 1.9 cm cavitary lesion with layering internal fluid within the posterior left lower lobe. There was an additional smaller cavitary lesion within the right upper lobe measuring 5 mm and a 1 cm ground glass nodule in the left lower lobe.

**Figure 2 tropicalmed-06-00207-f002:**
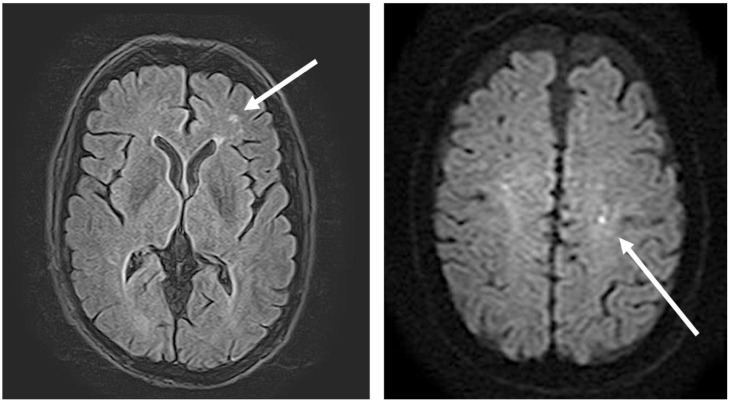
MRI of the brain with contrast. (**Left**): Scattered nonspecific supratentorial white matter FLAIR hyperintensities, suggesting chronic small vessel disease. (**Right**): Subcentimeter focus restricted diffusion left centrum semiovale extending to the subcortical white matter of the left precentral gyrus, concerning a recent small vessel infarct. No hemorrhage was present.

**Figure 3 tropicalmed-06-00207-f003:**
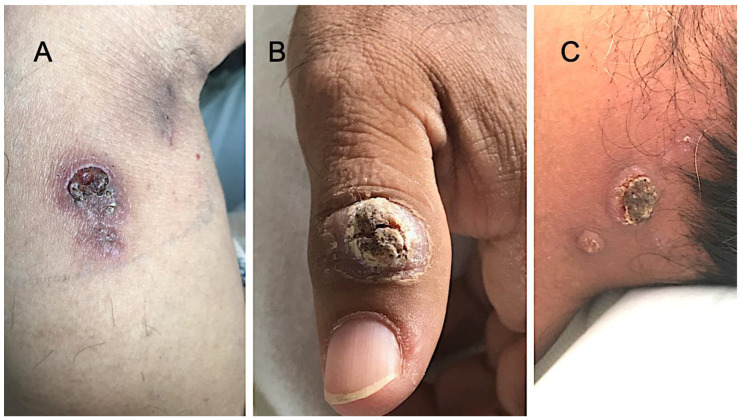
Skin lesions of the patient: (**A**) verrucous plaque of right forearm (the biopsy site); (**B**) large verrucous lesion of the left thumb; (**C**) nodular and verrucous lesions of the nape of the neck.

**Figure 4 tropicalmed-06-00207-f004:**
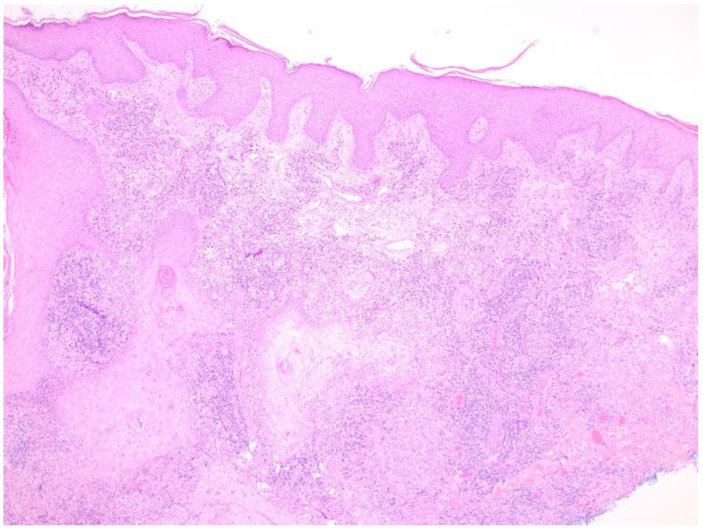
Photomicrograph of the biopsy specimen from the verrucous lesion on the right arm showing granulomatous dermatitis with pseudoepitheliomatous squamous proliferation, Hematoxylin and Eosin (H&E), 40×.

**Figure 5 tropicalmed-06-00207-f005:**
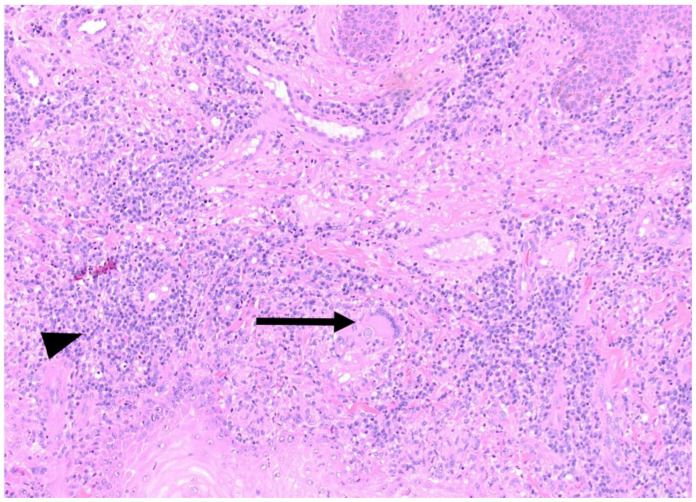
Photomicrograph of the biopsy specimen from the verrucous lesion on the right arm showing granulomatous inflammation (arrowhead) and multinucleated giant cell with internalized *Coccidioides* spherule (black arrow) (H&E), 100×.

**Figure 6 tropicalmed-06-00207-f006:**
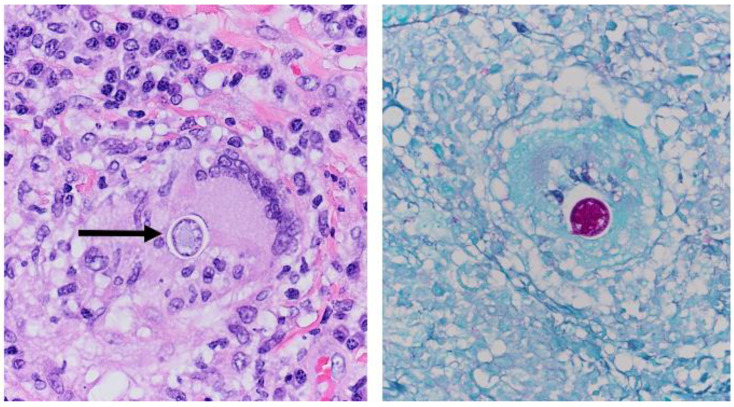
(**Left**): Increased magnification of Figure 5 showing multinucleated giant cell with internalized *Coccidioides* spherule (arrow) (H&E), 400×. (**Right**): Periodic acid-Schiff (PAS) stain for fungi showing *Coccidioides* spherule, 400×.

**Figure 7 tropicalmed-06-00207-f007:**
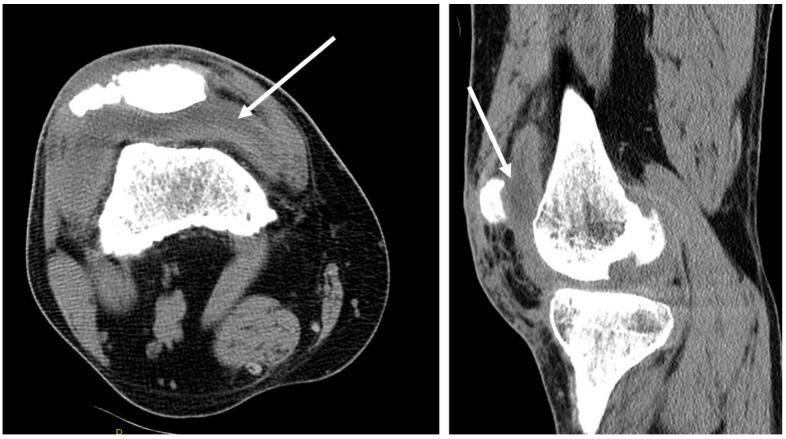
CT without contrast of R knee. Moderate suprapatellar joint effusion of right knee. (**Left**): Transverse view shows some extension of effusion laterally. (**Right**): Sagittal view shows some extension inferiorly.

**Figure 8 tropicalmed-06-00207-f008:**
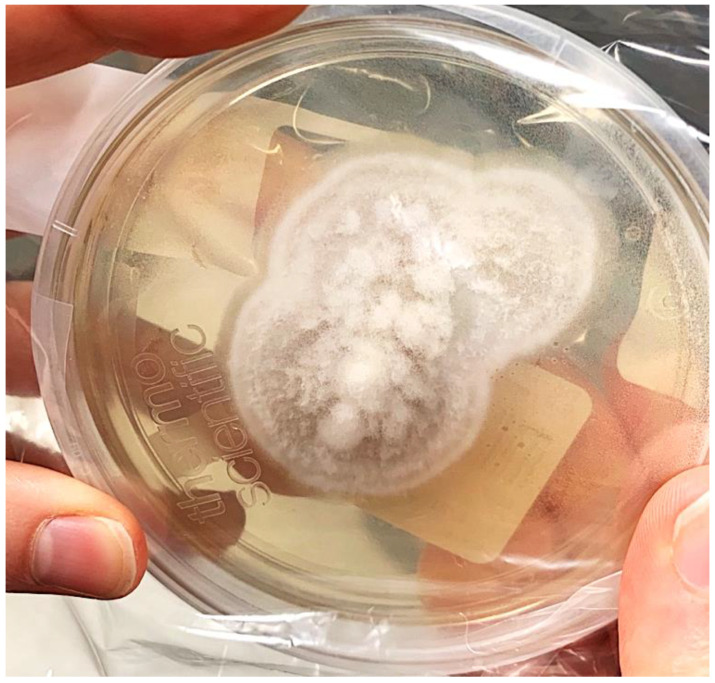
The typical white colonies of the mold form of *Coccidioides* on Sabouraud dextrose agar, which grew from the patient’s knee aspirate.

**Figure 9 tropicalmed-06-00207-f009:**
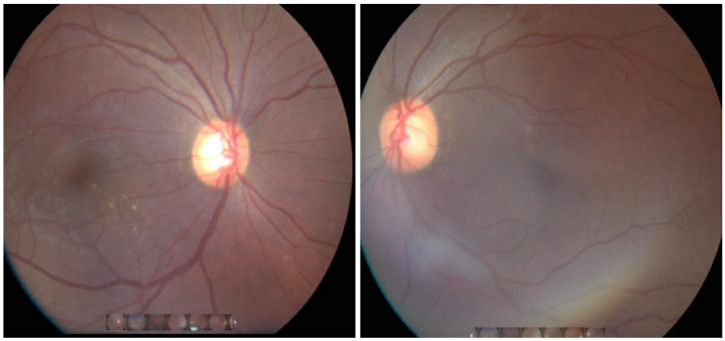
Fundoscopic findings. (**Left**): The right eye shows a few small circular flat creamy yellowish lesions along inferior and superior arcades, one lesion more pronounced along inferior temporal arcade, and a few pinpoint yellow lesions just inferior to fovea. (**Right**): The left eye showed a few circular creamy yellowish lesions along arcades, with one more prominent along the inferior arcade and few smaller circular, subtle yellow-white lesions just superior to the fovea.

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
