# Peer review of "Pandora’s Box: Disseminated Coccidioidomycosis Associated with Self-Medication with an Unregulated Potent Corticosteroid Acquired in Mexico"

_tropicalmed, 2021, doi:10.3390/tropicalmed6040207_

Round 1
Reviewer 1 Report
I do not have any suggestions for the authors.
This paper is an extensive and detailed presentation of a case of disseminated coccidioidomycosis in a diabetic patient who had self-medicated with a remedy containing corticosteroid. The diagnosis of his mycosis is based on firm findings. Each location of the disease is adequately documented and is discussed in comparison with the fiding observed in the consulted bibliography.
Author Response
No response necessary. I thank the reviewer for reading over the manuscript
Reviewer 2 Report
Ethical concern: Consent should be obtained for publication of patient information.
Format: The manuscript is long. In several ways both a case report and a review manuscript with text book information describing the disease and the epidemiology in detail. Consider making it shorter.
The manuscript ia both relevant and interesting. The paper is well written. The text is clear and easy to read. The conclusions are consistent with the arguments presented. The manuscript is very long. Most journals want manuscripts to be as stringent as short as possible. This manuscript contains a lot of information not necessary for presenting the case report itself. It is the choice of the journal how tough you want to be asking the authors to reduce the text and concentrate on the reason why they sent you this case report in the first place.
Author Response
The reviewer asked to shorten the manuscript but the editor has allowed the current length. I thank the reviewer for reading over the manuscript and I thank the editor for allowing the original length.